# Increasing Throughput in Warehouses: The Effect of Storage Reallocation and the Location of Input/Output Station

**Mohammed Alnahhal [1,*]**, **Bashir Salah [2]** and **Rafiq Ahmad [3]**

1. Mechanical and Industrial Engineering Department, American University of Ras Al Khaimah, Ras Al Khaimah P.O. Box 10021, United Arab Emirates
2. Department of Industrial Engineering, College of Engineering, King Saud University, P.O. Box 800, Riyadh 11421, Saudi Arabia; bsalah@ksu.edu.sa
3. Department of Mechanical Engineering, University of Alberta, Edmonton, AB T6G 1H9, Canada; rafiq.ahmad@ualberta.ca
* Correspondence: mohammed.alnahhal@aurak.ac.ae; Tel.: +971-7-246-8748

**Abstract:** Automatic warehousing systems are a sort of green technology that is becoming increasingly popular in the logistics business. Automated Storage/Retrieval Systems (AS/RS) are one of the most significant components of advanced automated logistics and manufacturing systems. The majority of AS/RS systems use input/output (I/O) points located in the lower left corner of the rack. These systems are reaching their maximum capacity because of their layout design limitations. Breakthrough solutions are needed to enhance the performance of existing systems. In this study, we examined how the location of I/O station can affect the total travel time. Another strategy for enhancement is a two-step preparation method. In this strategy, the allocation of the storage is changed, in the idle time, to be closer to the I/O point to reduce the service time for a class-based storage assignment. An analytical model was used to introduce for the first time optimal configurations of this strategy. We tested the suggested strategy using a simulation model created using R software, specifically designed for this purpose. Results showed that the two-step preparation strategy took between 1.2 and 1.9 h before the shift starts. The enhancement on throughput is almost the same for both possible locations of the I/O point. The results also showed that the two strategies (location of the I/O point and reallocation of storage) could increase throughput by about 21% to 28%, depending on parameters such as the number of orders and the height of the storage rack.

**Keywords:** automated storage and retrieval system; simulation; warehouses; material handling; throughput

## 1. Introduction

Generally, inventory management is very necessary to balance the supply with the demand using some optimization factors such as order quantity, reorder point, target stock and others [1]. However, this balance is not perfect because of the uncertainty in the system; therefore, large amounts of inventory are needed. Therefore, a fast way to store and handle this inventory is needed. Automated Storage/Retrieval Systems AS/RS are considered a green technology due to the importance of providing sustainability value to businesses to thrive in a sustainable environment. In recent years, the shortage of land resources has prompted a desire for continual AS/RS development based on information and automation. The tendency toward the use of automation will encourage the growth of warehouses with considerable height. This is especially significant in areas where land resources are scarce and land prices are skyrocketing [2]. Enhancing its throughput can occasionally lead to a reduction in the number of stacker cranes, lowering the amount of energy required, and lowering the system's total costs. With recent supply chain disruptions due to COVID-19 and political tensions, the importance of warehouses that handle massive amounts of products has become even more critical than before. This system is intertwined

with corporate material flow and is frequently used in industry [3]. Because of factors such as long processing times and high costs, manual processes cannot successfully foster the development of a sustainable economy. In this context, adopting sustainable green technologies, such as automated equipment for warehousing and order picking, has become a necessity in the promotion of sustainable social development, and green technology also allows for increased productivity and lower labor costs [4]. Increasing the efficiency of the system and improving its performance are the main goals behind most innovations in today's fast-changing business world, where being innovative is essential for staying ahead of the competition. In internal logistics, material handling is the act of moving, storing, protecting, and controlling materials within a facility [5]. Since its introduction, AS/RS technology has undergone many improvements. These systems are designed for storing and retrieving loads automatically in warehouses from a predefined storage location [6]. They reduce the time and costs associated with product damage and non-value-added labor in the process [7]. To improve the throughput performance of the system, a methodology must be developed to increase the efficiency of the AS/RSs.

### 1.1. AS/RS Components and Performance

Each AS/RS consists of one or more storage aisles that are serviced by a storage/retrieval (S/R) machine. Racks and aisles hold the materials stored in the storage area. AS/RS aisles have one or more input/output stations that deliver material into the storage system or remove it from the system. Ref. [8] illustrates the main components of the AS/RS system.

Due to the increased demand for manufactured goods, management has been motivated to find effective ways to boost productivity by considering the lack of necessary resources [9]. Increasing the performance of each component of the system would improve the productivity and efficiency of the system as a whole [10].

Several studies investigated issues related to the design and operation of AS/RS systems with the aim of optimizing their performance [11–13]. Models were proposed to describe the travel time for an (S/R) machine under randomized storage in AS/RS [14,15]. According to most traditional AS/RS, there are two main categories of operations: the dual command cycle (DCC) and the single command cycle (SCC). DCC involves both storage and retrieval, which follow one another in one route to decrease the distance traveled and, accordingly, the transaction time. The task of identifying this sequence is modeled as a traveling salesman problem involving multiple trips, with the objective of minimizing the total distance that the traveling salesman has traveled, and can be solved using a genetic algorithm [16]. Alternatively, SCC includes only one transaction per route, either storage or retrieval. The two modes of AS/RS operations are illustrated in Figure 1. The figure depicts the front view of the storage rack, and a random storage compartment for storage item (P) is shown while a different compartment for retrieval item (P′) is selected.

### 1.2. Enhancing Performance of AS/RS

Some studies concentrated on automation in warehouses as a green practice. In a study by Bartolini et al. [17], an evaluation model for the energy consumption and environmental impact of automation solutions of warehouses was proposed. They used literature review as the major methodology. They found that different storage locations and dwell-point strategies can affect energy efficiency in AS/RS. Generally, efficient use of AS/RS leads to shorter travel time and, therefore, lower energy consumption. Moreover, the research by Meneghetti et al. [18] investigated the best control policies for storage assignment and sequencing for both time and energy-based optimization. The mini-load AS/RS was investigated in a study by Lerher et al. [19], where an energy efficiency model was presented to reduce $CO_2$ emissions by using the best design. Shuttle systems were offered by some researchers as a better alternative than AS/RS for energy consumption, but the disadvantage of the shuttle system is the high investment cost. Furthermore, automated warehouses were considered with a low-carbon feature in a study by Li et al. [20], in which orders containing different items were batched together in an optimal way.

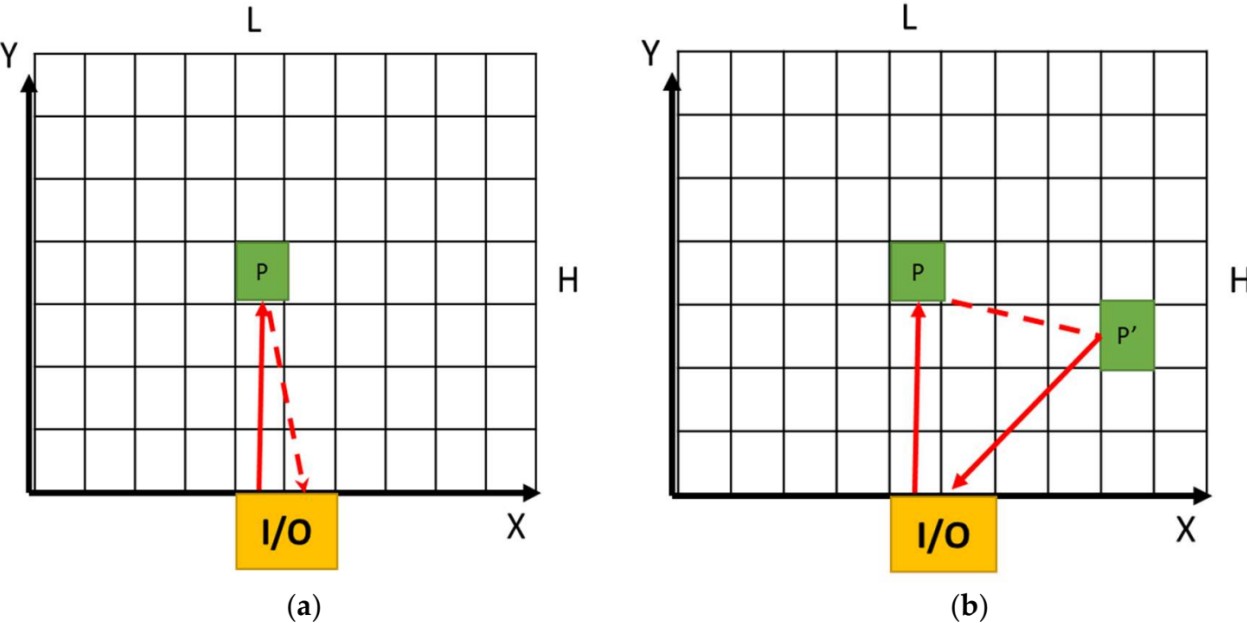

**Figure 1.** The two operation modes of AS/RS, (**a**) Single command cycle in case of storage, and (**b**) Dual command cycle.

Some studies investigated the optimal allocation of storage, such as the study by Wang et al. [21], who concentrated on flood control materials. Moreover, using mathematical programming, Bolaños Zuñiga et al. [22] investigated the optimization of the storage location assignment and the picker-routing problem. Four different scenarios were used. Other areas of research were also investigated, such as the study by Lewczuk et al. [23], who investigated energy consumption by distribution warehouses with systems such as AS/RS. Simulation was used in some studies, such as the article by Jerman et al. [24], where performance was analyzed. They discovered that designs with two separate I/O point locations outperform designs with a single I/O point location in terms of throughput. Medeiros et.al. [25] developed simulation models for mini-load systems. These models can be applied to find the average travel time of an (S/R) machine and the number of storage retrievals per cycle performed by a mini-load system [15]. The majority of the literature examines the performance of single-aisle AS/RSs with one I/O point [5]. An analysis of travel times for both SCC and DCC in different types of AS/RS configurations is an appropriate method of comparing control rules and storage assignment policies. This article proposes a reallocation algorithm to evaluate the performance of a stacker crane machine when it is idle with its I/O at the bottom of the rack and compares it with the situation in which the I/O station is in the corner. Using the position of the I/O station under the middle of the rack is known in the literature [26]. For example, Hao et al. [27] showed that shorter expected travel time results from this position, which leads to higher system throughput. We derived this algorithm using SCC and DCC, and class-based policy. It was used to assign a number of zones to the storage rack based on turnover rates. All items in the inventory system were ranked according to their contribution to the total demand, where class A items are the highest turnover products, class B items are the medium turnover products, and class C items are the lowest turnover products [28]. A simulation was conducted in order to determine the effectiveness of the reallocation scheme.

### 1.3. Problem Statement and Study Contribution

The storage allocation problem was considered in the literature assuming three situations: static allocation, dynamic allocation, and allocation during the idle time. An example of static allocation is the study by Yang et al. [29], where the storage load location was assigned based on its weight to get rack stability. An example of the dynamic allocation is

the study by Chen et al. [30], where the less-than-pallet load was relocated dynamically after finishing the previous retrieval order to enhance the system performance. However, such a dynamic strategy is only possible if the same pallet is retrieved at least two times. Little was published about the third type, in which the reallocation is performed in the idle time before the working shift. Enough time is usually available during performing this strategy; therefore, several movements are possible, and thus, considerable performance improvement is possible. The reallocation of storage in AS/RS given the current location of different products was generally overlooked in previous research. There are few studies about that, such as the one by Christofides and Colloff [28], who investigated the optimal way of rearranging items in a warehouse from their initial positions to their desired final locations. This is done when the stacker crane is idle. Heuristics were also developed in a study by Muralidharan et al. [31]. The efficiency and validity of the algorithm were illustrated through computer simulation. Moreover, a study by Carlo et al. [32] investigated the optimization of the rearrangement process in a dedicated warehouse, given the current location of products. However, such studies used complicated mathematical models and heuristics. In our paper, we investigated a simpler way to make the rearrangement, and the optimal configuration for this rearrangement process was found. Then, simulation was used to validate the idea. A similar study is the one by Salah et al. [6], who investigated a two-step method to increase the throughput and to decrease the service time per order when the I/O station is on the point (0, 0). The proposed idea by Salah et al. [6] needs, however, more investigation about the system configurations to further increase the throughput. Therefore, this study further investigates the two-step method in the case where the I/O station is under the middle of the storage rack and compares the results for the two proposed positions of the I/O station. Increasing the throughput can sometimes lead to reducing the needed number of the very expensive stacker cranes. To make the most of this strategy's potential, another contribution is that this study investigates more deeply the best size of the evacuation area, in which the evacuation process and reallocation process occur in two steps. The best configuration is done using an analytical model. The simulation model was built using R software, which is an open-source and flexible software that can be used for many purposes and can be customized to simulate complex systems. To the best of the authors' knowledge, this is the first study with such objectives that investigates the system configurations in depth. To summarize, the main hypothesis in this study was that reallocating storage using the proposed approach can improve system performance when the I/O station is in the left corner or under the middle of the rack. Simulation results were used to test the hypothesis. Several trials of simulation results were obtained and averaged to allow for a more accurate comparison of scenarios with and without storage reallocation.

## 2. Methodology

This study considered a two-step method to decrease the service time in the AS/RS system by making reallocation of the tote bins before the working shift starts. The reallocation process was expected to make the tote bins needed in the next few hours to be closer to the I/O station. Two steps were considered for that:

1. Evacuation Step: evacuating the storage locations of tote bins that will not be needed in the next few hours. Only some of the A-items are evacuated. The B and C items are already far from the I/O station;
2. Gathering Step: reallocation of the expected needed tote bins that are far from the I/O to be closer in the evacuated area near to the I/O point. In this step, all types of items (A, B, and C) can be reallocated to the near area (the area originally dedicated for A-items).

The effect of this strategy was investigated using simulation in two different cases:

1. Case 1: the I/O station is in the location (0, 0) (lower left-hand corner of the storage rack);

2. Case 2: the I/O station is just under the middle of the storage rack. The distance between the center of the I/O stations and the lowest line of the storage rack is h/2, where h is the cell height. The cell in this paper is the storage compartment in the storage rack (storage location).

Therefore, four different scenarios were investigated. However, the second case (with its two scenarios) was the main focus of this study. For each scenario, different averages of service times and throughput were expected. The following assumptions were needed:

- The exact demand for the next few hours is known in the idle time before the shift starts;
- A single rack is used;
- The I/O station can be at the point (0, 0) or under the middle of the storage rack;
- A tote bin in a storage order at the beginning of the day can be later needed as a retrieval order on the same day;
- There is enough time for the two-step storage reallocation process before the working shift starts;
- The classed-based (ABC) assignment of the storage rack is used to allocate storage locations on the rack.

The following steps summarize the method of the paper (see Figure 2). Details for these steps, especially the calculations, were given afterward:

1. Set the parameters of the storage rack, locations of tote bins, and I/O station's position;
2. Calculate the service time for each cell;
3. Make cells numbering from the cells with the lowest service time to the highest service time;
4. Read the forecasted demand (assumed to be accurate);
5. Determine the locations of the tote bins needed according to the forecasted demand
6. Evacuation Step:

    a. Determine the size of the evacuation area;
    b. Determine the number of evacuated cells;
    c. Set the equation to calculate the reallocation time;
    d. Evacuation process;

7. Gathering Step:

    a. Find the number and locations of the kept empty locations;
    b. Reallocation of far and needed tote bins to be closer to the I/O station;

8. Shift work: storage and retrieval orders are processed;
9. Collect statistics.

Table 1 summarizes the initial parameters needed for the first step mentioned above. According to the table, 70% of the cells will be occupied at the beginning of the shift. This occupation was assumed to be random. Moreover, 50% of the area of the rack will be for the A-items (tote bins). These tote bins contribute to 80% of the whole orders demand. The tote bins that need to be considered for reallocation are those in cells with numbers more than the size of the evacuation area, as will be explained later. The stacker crane will be busy 90% of the time. In the beginning, during development of the simulation model, the number of orders (SCC + DCC) was chosen to be 600 orders. But later, this number was changed to get valuable results about the effect of changing it from 600 to 500, and then to 400.

A study by Salah et al. [6] concentrated on the case in which the I/O station is on the point (0, 0). The reader might refer to that paper to find the equations for the service time per order. In this study, a comparison of the two cases of different I/O station locations was investigated. The new location for the I/O station is below the middle of the storage rack. Figure 3 shows the two allocations of the class-based systems when L = 50 and H = 30.

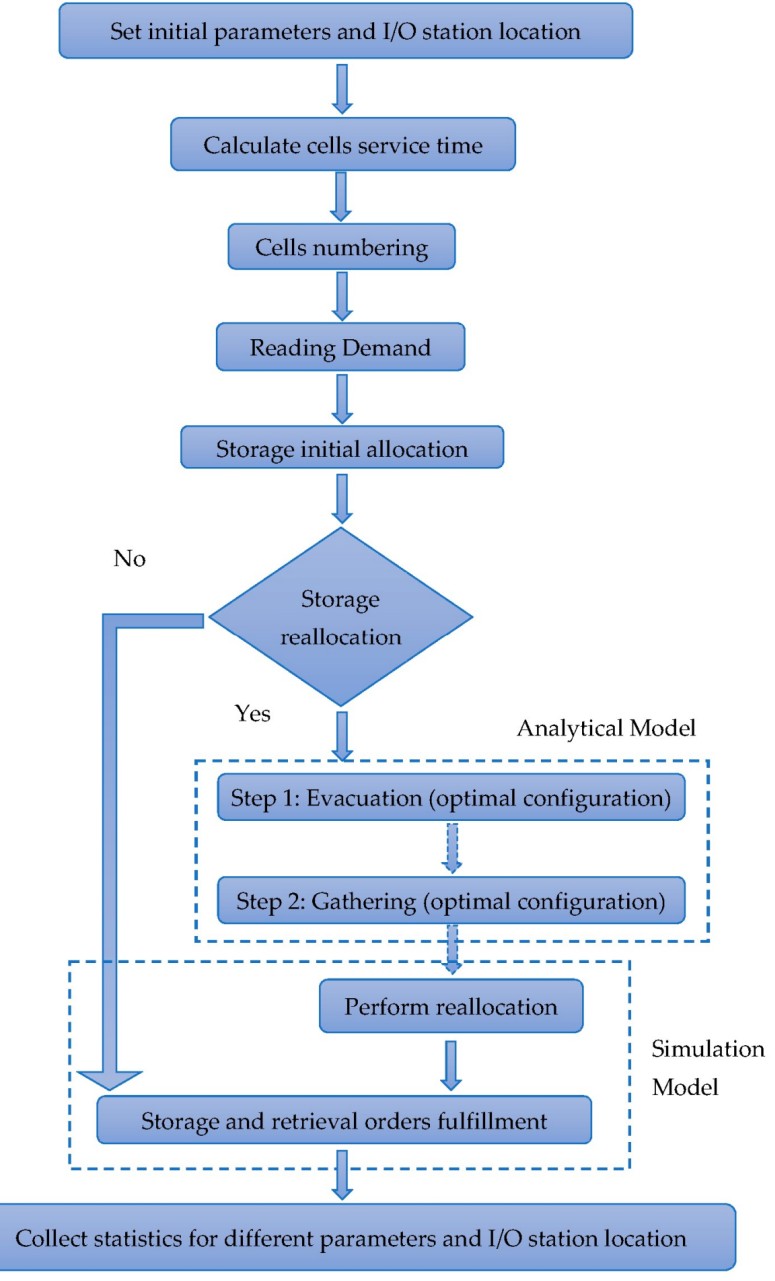

**Figure 2.** Study Methodology.

**Table 1.** Initial values of study parameters.

| Parameter | Value | Parameter | Value |
|---|---|---|---|
| Rack utilization ($U_R$) | 70% | Average % retrieval orders (R) | 25% |
| Number of horizontal cells (L) | 50 | Stacker-Crane horizontal speed ($v_h$) | 5 m/s |
| Number of vertical cells (H) | 30 | Stacker-Crane vertical speed ($v_h$) | 2 m/s |
| Cell length (l) | 0.6 m | ABC allocation for cells | A = 50%, B = 30%, C = 20% |
| Cell height (h) | 0.4 m | ABC allocation for Demand (of orders) | A = 80%, B = 15%, C = 5% |
| Average % DCC | 50% | Storage retrieval time utilization ($U_{sr}$) | 90% (stacker crane in use) |
| Average % storage orders (S) | 25% | I/O station | Below the middle of the rack or on the point (0, 0) |

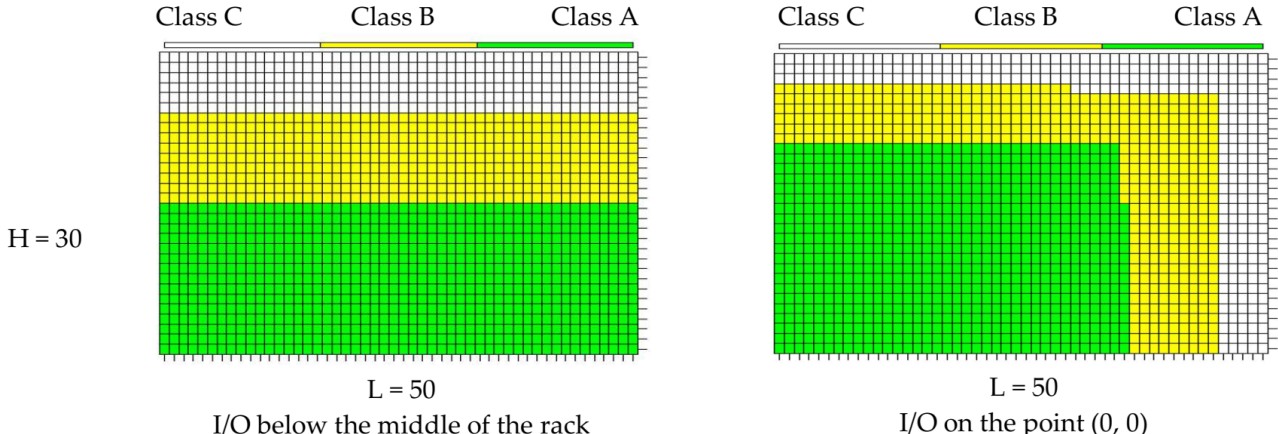

**Figure 3.** Storage allocation for the two different positions of I/O station.

When the I/O station is below the middle of the rack, *SCC* service time can be found using Equation (1):

$$SCC\ time = 2\max\left(\frac{\left|x_j - 0.5 - L/2\right|l}{v_h}, \frac{y_j h}{v_v}\right) + 2T_{pd} \tag{1}$$

where $x_j$ and $y_j$ are the Cartesian locations for the cell. The variables ($v_h$, $v_v$, $l$, and $h$) are in Table 1. $T_{pd}$ is pickup-and-deposit time (loading/unloading time), and it was set to be 2.5 s. *DCC* service time can be found using the following equation:

$$DCC\ time = \max\left(\frac{\left|x_{j1} - 0.5 - L/2\right|l}{v_h}, \frac{y_{j1} h}{v_v}\right) + \max\left(\frac{\left|x_{j1} - x_{j2}\right|l}{v_h}, \frac{\left|y_{j1} - y_{j2}\right|h}{v_v}\right)$$
$$+\max\left(\frac{\left|x_{j2} - 0.5 - L/2\right|l}{v_h}, \frac{y_{j2} h}{v_v}\right) + 4T_{pd} \tag{2}$$

The horizontal and vertical speeds ($v_h$, $v_v$) are different. Therefore, in the previous two equations, the maximum time in any direction is the one used. The evacuation time (RT) in the evacuation step can be measured using Equation (3), where the dwell point is the last position of the previous movement:

$$ET = \max\left(\frac{\left|x_{(j-1)2} - x_{j1}\right|l}{v_h}, \frac{\left|y_{(j-1)2} - y_{j1}\right|h}{v_v}\right) + \max\left(\frac{\left|x_{j1} - x_{j2}\right|l}{v_h}, \frac{\left|y_{j1} - y_{j2}\right|h}{v_v}\right) + 4T_{pd} \tag{3}$$

where $x_{j1}$ and $y_{j1}$ are the coordinates for the cell that requires evacuation, and $x_{j2}$ and $y_{j2}$ are the coordinates for the suggested cell to receive the tote bin. $x_{(j-1)2}$ and $y_{(j-1)2}$ are the coordinates for the previous movement's last step.

To find the number of evacuated cells near the I/O station, we should consider the total number of retrieval orders (R) = the total number of storage orders (S). Sometimes even if the R = S, the number of storage orders at a certain point in time is more than the number of retrieval orders to that point in time (because of randomness). This means that there must be enough space, in the evacuation area, for this difference. To investigate this, it is known in statistics that the probability of getting exactly *k* successes in *n* independent Bernoulli trials is given by the probability mass function (pmf):

$$f(k, n, p) = P(X = k) = \binom{n}{k} p^k (1-p)^{n-k} \tag{4}$$

where $n$ can be considered as the total number of orders, $k$ is the number of storage/retrieval orders, and $p$ is the probability that the order is a storage/retrieval one. The concentration here is on the SCC. In DCC, however, two orders (storage and retrieval) occur at the same time, and that means that the effect of one order cancels the effect of the other one. The cumulative distribution function (CDF) can be expressed as follows:

$$F(k, n, p) = P(X \le k) = \sum_{i=0}^{\lfloor k \rfloor} \binom{n}{i} p^i (1-p)^{n-i} \tag{5}$$

If R = S ($p$ = 0.5), the probabilities that the number of storage orders is more than the number of retrieval orders for the first 9, 10, and 11 orders are shown in Table 2 based on Equation (4). When the total number of orders is 9, the probability that this difference is 5 is about 7% ($P(X = 2)$). The probability to get a difference of 7 or more ($P(X \le 1)$) is less than 2% (0.01758 + 0.00195). In other words, the probability to get a difference of more than 5 is 2% (less than 5%). Suppose that the concentration is on the maximum difference between storage orders and retrieval orders, after which this probability is still less than 5% (MDOP5). In other words, MDOP5 is the maximum difference $d$ that still satisfies the following formula:

$$P\left(X \le \left\lceil \frac{n}{2} - j \right\rceil\right) \ge 0.05 \tag{6}$$

where $j = \frac{d_j + (n \bmod 2)}{2}$ and $\lceil x \rceil$ is the round to the upper integer of $x$
The formula in Equation (6) can be rewritten as follows:

$$P\left(X \le \left\lceil \frac{n}{2} - \frac{d_j + (n \bmod 2)}{2} \right\rceil\right) \ge 0.05 \tag{7}$$

In Table 2, MDOP5 = 5, 6, and 7 for the cases when the number of orders ($n$) is 9, 10, and 11, respectively.

**Table 2.** The probability that storage is more than retrieval orders for the first 9, 10, and 11 orders.

| j | **$n = 9$** Storage Orders − Retrieval Orders | Probability | **$n = 10$** Storage Orders − Retrieval Orders | Probability | **$n = 11$** Storage Orders − Retrieval Orders | Probability |
|---|---|---|---|---|---|---|
| 1 | 5 − 4 = 1 | 0.24609 | 64 = 2 | 0.20508 | 6 − 5 = 1 | 0.22559 |
| 2 | 6 − 3 = 3 | 0.16406 | 7 − 3 = 4 | 0.11719 | 7 − 4 = 3 | 0.16113 |
| **3** | **7 − 2 = 5** | **0.07031** | **8 − 2 = 6** | **0.04395** | 8 − 3 = 5 | 0.08057 |
| 4 | 8 − 1 = 7 | 0.01758 | 9 − 1 = 8 | 0.00977 | **9 − 2 = 7** | **0.02686** |
| 5 | 9 − 0 = 9 | 0.00195 | 10 − 0 = 10 | 0.00098 | 10 − 1 = 9 | 0.00537 |
| 6 | - | - | - | - | 11 − 0 = 11 | 0.00049 |
| | Sum | 0.5 | Sum | 0.37695 | Sum | 0.5 |

Figure 4 shows the MDOP5 values for the first 450 orders when S = R. According to the figure, there is no need to keep empty space (cells) besides the I/O station more than 36 storage locations. However, these 36 storage locations are needed only at the end of working hours for these 450 orders. Only part of the 36 storage locations will be needed most of the time.

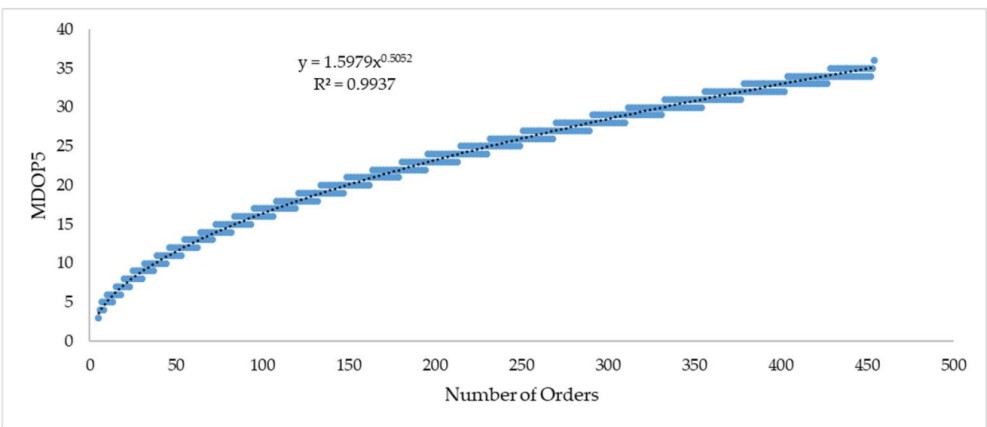

**Figure 4.** The MDOP5 values for the first 450 orders.

To account for that, Figure 5 shows the average values of the probability (P) that the difference between the number of storage and retrieval orders (storage orders are greater than retrieval orders) is *d* or more. In this case, a number of evacuated cells of *d* should be maintained empty. The figure shows that if all the orders (storage and retrieval) are 450, 10 empty compartments are enough to cover this difference 80% of the time. If there is not enough space for a certain storage order, it is stored outside the evacuation area.

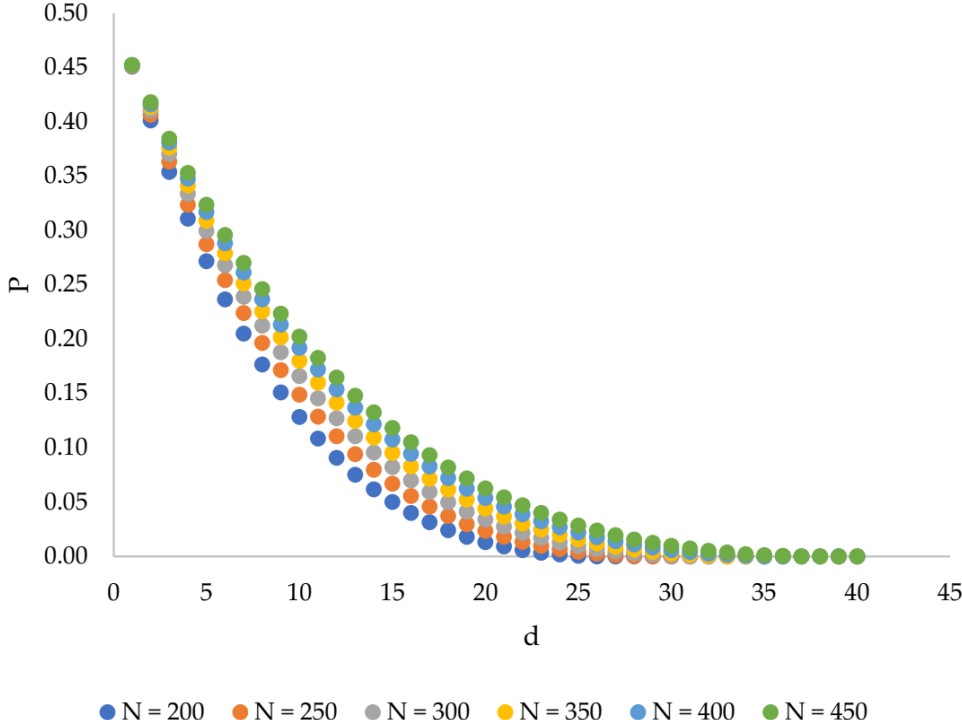

**Figure 5.** Relationship between *P* and *d*.

Different values are obtained for different Ns. The y-axis in Figure 5 is the average value of $P_{nj}$, where $P_{nj}$ is the probability of a difference $d_j$ when the total number of orders is *n*:

$$P = \frac{\sum_{n=2}^{N} P_{nj}(d \geq d_j)}{N - 1} \qquad (8)$$

If the concentration is on the 5%, according to Figure 5, the following relationship can approximate the value of the difference $d$ for a given $N$ value:

$$d = 0.023\,N + 10.9 \tag{9}$$

It means that there is a 5% chance that the number of storage orders is more than the number of retrieval orders by $d$. The area near the I/O station (evacuation area) should be enough to cover all the retrieval orders with known storage locations at the beginning of the shift plus the difference $d$. This area's size is defined by the last cell number (*LE*) in the evacuation area, which can be found by the following formula:

$$LE = IR + 0.8\,(0.023(R+S) + 10.9) \tag{10}$$

where *IR* is the number of retrieval orders for tote bins already available at the beginning of the shift. As explained earlier, a stored tote bin at the beginning of the day can be later retrieved on the same day. The number of excluded cells should be the last term of Equation (10), which is $0.8\,(0.023(R+S) + 10.9)$. For example, if $R + S$ is 300, then 14 excluded cells should be kept empty in the gathering step. These 14 cells can be determined randomly in the near area of the I/O station, or, for example, the first 14 empty cells with odd numbers. The multiplication of 0.8 in the equation is because the concentration is on the A-class zone assumed to contribute to about 80% of the orders.

Just before the initial evacuation step, the cells in the evacuation area can be classified as follows:

1. Empty cells and their average number equals $LE\,(1 - U_R)$;
2. Cells full of the tote bins demanded during the next few hours. The number of these cells can be written as *NIR* (*NIR* are the *IR* that are located in the area near the I/O station);
3. Cells full of tote bins that are not required in the next few hours. These cells should be evacuated in the first evacuation step. On average, the number of these cells is *EC*:

$$EC = LE\,U_R - NIR \tag{11}$$

This number of evacuated cells, however, might not be possible to be evacuated if there is not enough space in the A-area around the evacuation area to evacuate all these cells. The number of empty cells in the area around the evacuation area (area 2) is *ECA2*, assuming that A-zone (area 1 and area 2) is 50% of the whole rack:

$$ECA2 = (1 - U_R)\left(\frac{HL}{2} - LE\right) \tag{12}$$

Therefore, the total theoretical number of evacuated cells (*NEC*) will be the minimum of *EC* and *ECA2*:

$$NEC = \min\left(LE\,U_R - NIR,\ (1 - U_R)\left(\frac{HL}{2} - LE\right)\right) \tag{13}$$

Figure 6 shows three areas: area 1 near the I/O station, area 2 around area 1, and area 3, which is the rest of the storage rack. Area 1 is defined by LE. Area 2 contains cells with cell numbers between LE and LH/2. The third area is the rest of the rack, which contains B and C zones.

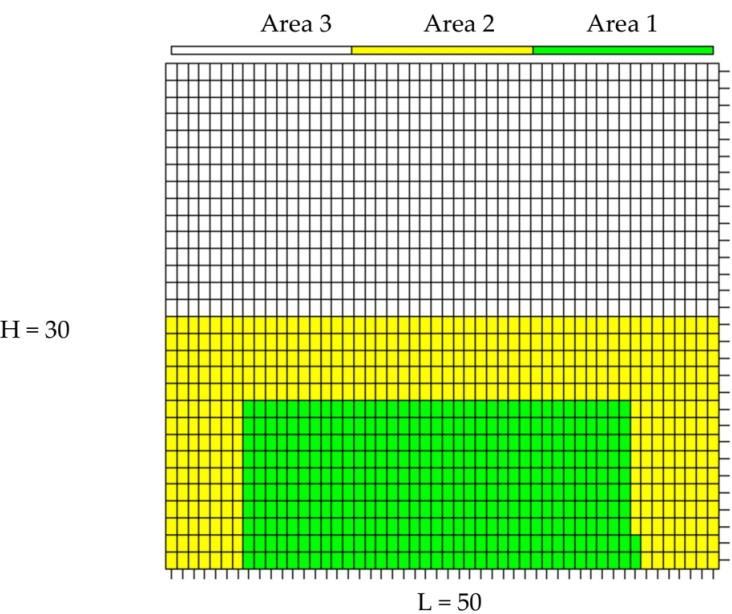

**Figure 6.** Three areas on the storage rack.

The simulation model takes the *LE* as an input, and it provides the number of evacuated cells as output. The difference between the theoretical one and *NEC* depends on real rack utilization in each area. This difference can be considered as one way to verify the results of the simulation model. In the case that $EC \leq ECA2$, the number of reallocated cells to area 1 from the rest of the storage rack (in the gathering step) will be *NRC*:

$$NRC = LE - NIR \tag{14}$$

In this case, the near area (evacuation area) will be filled in with the tote bins at the end of the gathering step. However, in case the geometry of the figure plays a role where $EC > ECA2$, then the formula to find the number of reallocation movements will be

$$NRC = LE - NIR - (EC - ECA2) \tag{15}$$

The number of movements of the stacker crane in the second step (gathering step) will be higher than the number of movements in the first step. This is because the start of the first evacuation step happens when there are $1 - U_R$ empty cells, while at the end of the second (gathering) step, area 1 will be full or almost full. So, the difference between the number of movements in the two steps can be approximated to be about $(1 - U_R)\,LE$. Therefore, it is expected that the time of the gathering step is larger than the time of the evacuation step. Another reason is that the evacuation step movement is inside the A-zone on the storage rack. However, the gathering step movements are from the whole storage rack to area 1. In other words, the average travelled distance per order of the gathering step is larger than the average travelled distance in the evacuation step.

One important point here is that reallocation movements should only be made if the difference in cell numbers indicates that they save a significant amount of time. Therefore, the reallocation process in the gathering step is started from the closest cell in area 2 to the closest cell to the I/O station in area 1.

## 3. Results and Analysis

Figure 7 shows the initial allocation of storage, the first step (evacuation), the second step (gathering), and the final allocation of storage items. The first three phases are before the working shift starts. The last one is the first few hours of the shift (according to the determined number of orders).

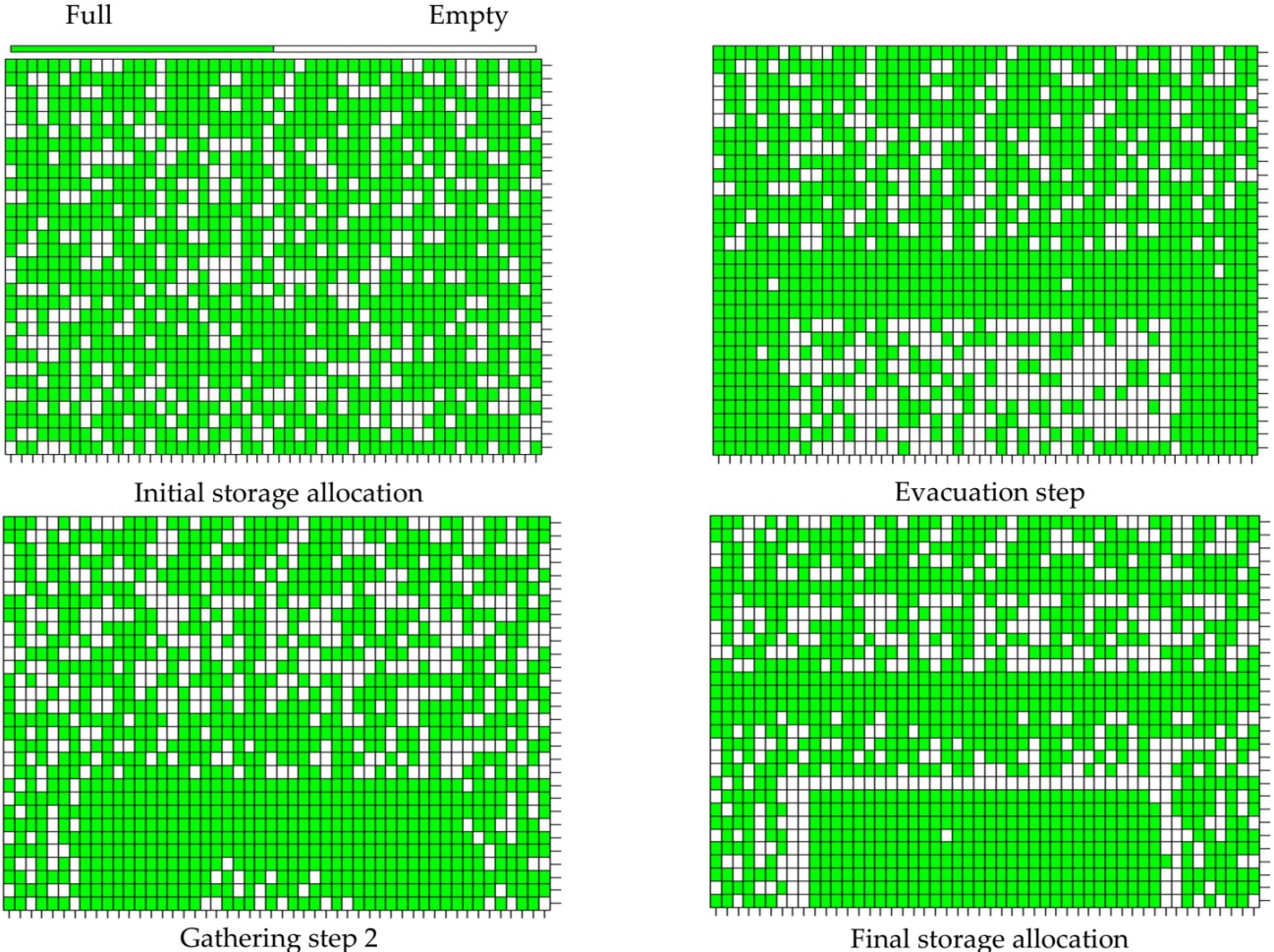

**Figure 7.** The four phases of storage allocation in case 2.

Results were obtained from the simulation model where different scenarios were tried by changing the following factors:

1. The number of orders (400, 500, and 600);
2. The height of the storage rack (30 and 40);
3. The location of the I/O station (case 1: at the left down corner with coordinates of (0, 0), and case 2: under the middle of the storage rack);
4. The two-step method (use it or not).

To get dependable results, five trials were used for each case. Table 3 shows the averages of results for the storage rack of H = 30. ST is the average service time in hours, and TP is the average throughput in orders per hour. As expected, a lot of time savings were obtained from changing the location of the I/O station and from the two-step method. The savings from the two-step preparation method was even better than changing the I/O station. Nevertheless, merging both strategies together can save even more. The throughput was calculated based on the assumed fixed utilization of the stacker crane of 90% in all the scenarios.

Table 4 shows the results for the case with H = 40. In the two tables, the savings in service time is still lower than the time consumed in the two-step method. However, the main assumption here is that the two preparation steps are before the working shift starts. The height of the storage rack affected the results. To have a better visualization of this effect, Figure 8 shows the difference between the results of throughput with H = 30 and H = 40.

**Table 3.** Results of different scenarios with (H = 30).

| Number of Orders | I/O Station Is (0, 0) | | I/O Station Is below the Middle of Storage Rack | |
|---|---|---|---|---|
| | **Scenario 1: Without the Two Steps** | **Scenario 2: With the Two Steps** | **Scenario 3: Without the Two Steps** | **Scenario 4: With the Two Steps** |
| 400 | ST = 1.72 h<br>TP = 232.00 orders/h | ST = 1.50 h<br>TP = 267.34 orders/h<br>Step 1 time = 0.47 h<br>Step 2 time = 0.77 h | ST = 1.57 h<br>TP = 254.34 orders/h | ST = 1.38 h<br>TP = 289.87 orders/h<br>Step 1 time = 0.45 h<br>Step 2 time = 0.74 h |
| 500 | ST = 2.15 h<br>TP = 231.92 orders/h | ST = 1.91 h<br>TP = 262.08 orders/h<br>Step 1 time = 0.48 h<br>Step 2 time = 0.86 h | ST = 2.00 h<br>TP = 249.50 orders/h | ST = 1.78 h<br>TP = 280.36 orders/h<br>Step 1 time = 0.46 h<br>Step 2 time = 0.88 h |
| 600 | ST = 2.59 h<br>TP = 231.59 orders/h | ST = 2.32 h<br>TP = 258.29 orders/h<br>Step 1 time = 0.48 h<br>Step 2 time = 0.90 h | ST = 2.4 h<br>TP = 250.53 orders/h | ST = 2.14 h<br>TP = 280.02 orders/h<br>Step 1 time = 0.53 h<br>Step 2 time = 0.91 h |

**Table 4.** Results of different scenarios with (H = 40).

| Number of Orders | I/O Station Is (0, 0) | | I/O Station is below the Middle of Storage Rack | |
|---|---|---|---|---|
| | **Scenario 1: Without the Two Steps** | **Scenario 2: With the Two Steps** | **Scenario 3: Without the Two Steps** | **Scenario 4: With the Two Steps** |
| 400 | ST = 1.86 h<br>TP = 215.21 orders/h | ST = 1.53 h<br>TP = 261.70 orders/h<br>Step 1 time = 0.58 h<br>Step 2 time = 0.93 h | ST = 1.74 h<br>TP = 229.65 orders/h | ST = 1.45 h<br>TP = 275.79 orders/h<br>Step 1 time = 0.59 h<br>Step 2 time = 0.94 h |
| 500 | ST = 2.29 h<br>TP = 217.90 orders/h | ST = 1.92 h<br>TP = 260.03 orders/h<br>Step 1 time = 0.67 h<br>Step 2 time = 1.05 h | ST = 2.18 h<br>TP = 229.31 orders/h | ST = 1.85 h<br>TP = 269.69 orders/h<br>Step 1 time = 0.63 h<br>Step 2 time = 1.06 h |
| 600 | ST = 2.78 h<br>TP = 215.91 orders/h | ST = 2.36 h<br>TP = 254.02 orders/h<br>Step 1 time = 0.67 h<br>Step 2 time = 1.18 h | ST = 2.59 h<br>TP = 231.26 orders/h | ST = 2.25 h<br>TP = 267.14 orders/h<br>Step 1 time = 0.64 h<br>Step 2 time = 1.17 h |

Figure 8 shows a bigger jump in throughput when the two-step method was used when H = 40. It also shows that the effect of changing the location of the I/O station is still valid but with a lower value. The enhancement in the throughput when H = 30 is from 21% to 25% if both enhancement strategies are combined (changing the I/O station location and the two-step method). On the other hand, this enhancement is about 24% to 28% when H = 40. The ranges here are because of the different number of orders, where better results are obtained for a lower number of orders.

To show the effect on the savings in the service time, Figure 9 shows the service time percentage compared to the base scenario (case 1 and without the two steps). This base scenario will always have a 100% value. The effect of the two-step method was higher than the effect of changing the location of the I/O station. When both strategies are combined, the percentage of the service time can be 78% of the base scenario when the number of orders is 400. There is some effect for the number of orders in the two-step method, where a lower number of orders gives better results. However, this effect is not significant for changing the location of the I/O station.

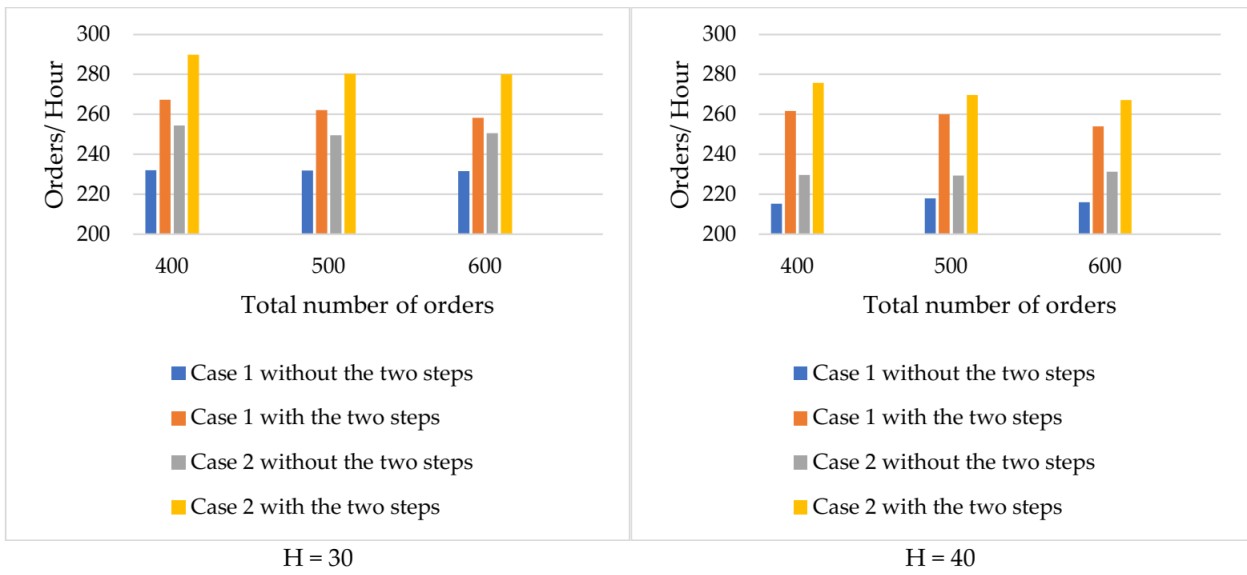

**Figure 8.** The effect of H on the throughput.

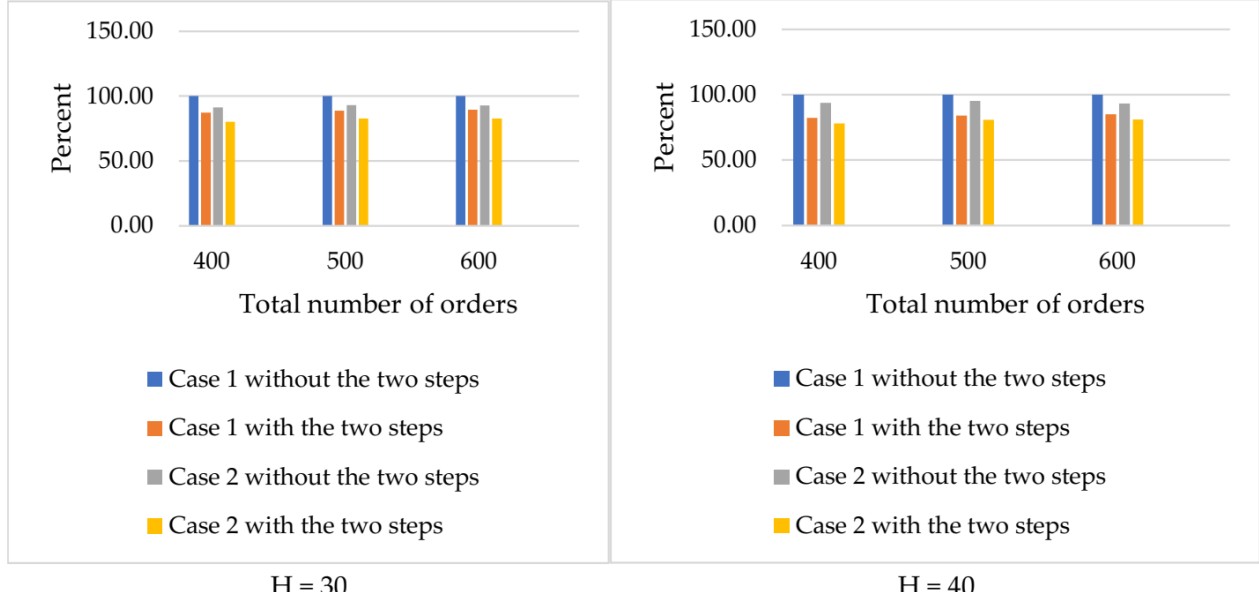

**Figure 9.** The service time percentage compared to the base scenario.

Even though the two enhancement strategies reduced the service time by about 22%, the preparation (two-step) method consumed some energy. Since the throughput of the system was enhanced to about 21% to 28%, the proposed enhancement strategies are only justifiable if higher throughput of the AS/RS system is required. The demand for higher throughput can be stable or have some flexibility to react to an increase in the demand, especially if the just-in-time philosophy is used.

The enhancement in the throughput and service time found in this study is in agreement with the study by Salah et al. [6], which concentrated on the two-step method. However, extra enhancement can be found in this study as a result of combining the two strategies together (two-step method and changing the location of the I/O station). This was done for the first time in this study.

## 4. Conclusions

This study investigated two strategies to enhance the throughput and service time of the AS/RS system, which are the best location for the I/O station and the two-step method to reallocate the tote bins before the shift starts. In the first strategy, two locations were investigated, which are the (0, 0) and under the middle of the storage rack. In the second strategy, the two-step (evacuation and gathering) method was investigated. A simulation model using R software was built to analyze the system and obtain the results. Analytical equations were used to find the best size for the evacuation area. Equations were also utilized to find the number of movements in the two steps to verify the results of the simulation. The equations provided are suitable for the two different positions of the I/O station. Results showed that combining the two strategies can enhance the throughput of the system by up to 28%, depending on the storage rack size and the number of orders. The value of this study is of great interest for decision makers in the case that the throughput of the system is required to be enhanced. The reallocation problem reduces the travel time during the working shift and, therefore, reduces the energy consumption during the working shift. Better throughput might lead sometimes to a lower number of stacker cranes and, therefore, lower consumption of energy and lower total costs of the system. The investigation, however, considered mainly the case that the number of the storage and retrieval orders are the same or almost the same during the shift. Some modifications to the formulas can be checked in future research if different numbers of storage and retrieval orders are there. Another limitation of the study is the assumption of a single rack. The number of bins in one storage compartment was assumed to be one. In some systems, however, two bins can be used to enhance the space utilization. Future research can also investigate the reallocation problem for a double-deep racking system. According to this current research, the storage retrieval system consists of multiple aisles. Within each aisle, there is an S/R machine to serve the front rack. Future research can investigate the reallocation problem in a three-dimensional structure. If the current capacity is enough and no more enhancement is needed, then the two-step method should be reconsidered because of the extra energy consumption it makes in idle time. Furthermore, the shape factor can also be further investigated in the future.

**Author Contributions:** Conceptualization, M.A. and B.S.; methodology, M.A. and B.S.; software, M.A.; validation, M.A. and B.S.; formal analysis, M.A.; investigation M.A. and B.S.; resources, M.A., R.A. and B.S. data curation, M.A. and B.S.; writing—original draft preparation, B.S. and M.A.; writing—review and editing, M.A.; and B.S.; visualization, M.A and R.A..; supervision, B.S. and R.A.; project administration, M.A and R.A.; funding acquisition, B.S. All authors have read and agreed to the published version of the manuscript.

**Funding:** This study received funding from King Saud University, Saudi Arabia through researchers supporting project number (RSP-2021/145). Additionally, the APCs were funded by King Saud University, Saudi Arabia through researchers supporting project number (RSP-2021/145).

**Institutional Review Board Statement:** Not applicable.

**Informed Consent Statement:** Not applicable.

**Data Availability Statement:** Not applicable.

**Conflicts of Interest:** The authors declare no conflict of interest.

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
