# Peer review of "Increasing Throughput in Warehouses: The Effect of Storage Reallocation and the Location of Input/Output Station"

_sustainability, doi:10.3390/su14084611_

Round 1
Reviewer 1 Report
Thank you for the opportunity to review this interesting article. I took interest and pleasure to read this paper. This study investigated two strategies to enhance the throughput and service time of the AS/RS system, which are the best location for the I/O station and the two-step method to reallocate the tote bins before the shift starts. Below are my comments:
Significance:
- The scientific content of this paper is correct for me and deserves to be published.
- Please clarify the hypotheses test used in this paper
- The research gap should be emphasized in the literature review section. I would also kindly ask to cite the very relevant research paper on inventory policies which play an important role in warehouse operation, to increase the quality of the work
https://doi.org/10.1016/j.tre.2021.102508
https://doi.org/10.3390/math8081210
https://doi.org/10.1016/j.clscn.2022.100042
Scientific soundness:
- The subject addressed in this paper is relevant.
- The study has been correctly designed and is technically sound.
Overall evaluation:
- The English language quality of this paper is globally appropriate and acceptable. However, some minor revisions and spell check seem to be necessary.
As a conclusion, my suggestion to the editor is to accept this paper for publication after minor revision.
Reviewer 2 Report
The paper deals with the detailed problem of rationalizing the transport cycle of the stacker crane in AS/RS taking into account the position of the I/O point and the possibility of relocating units during the idle time of the stacker crane to increase the throughput of the system.
The article presents a precise cause-and-effect sequence, a well-described methodology, and assumptions of the method. The obtained results are valuable, and experimental data are selected appropriately (but strictly for the experiment). It shows the usefulness of the method and its potential application to the efficiency of logistics processes.
The article is new, written in an appropriate language (both in terms of specialist and stylistic vocabulary), and may be published in Sustainability due to the topic; however, authors should consider the following comments:
1. The authors state that AS/RS is considered a green technology and make it one of the basic assumptions of the article. Still, they do not provide a sufficiently strong confirmation of it in the literature review. In the conclusions, this statement is also not emphasized enough (it is essential from the point of view of the journal's profile).
2. The statement: "In recent years, the shortage of land resources has prompted a desire for continual AS/RS development based on information and automation" requires additional elaboration.
3. The authors state that: "Since its introduction at the turn of the centuries ..." meanwhile, stacker cranes and their systems were used already in the 1980s and probably early.
4. Figure 2 is not precise. It is unknown whether the drawing shows the front of the rack zone or the side projection of one rack wall?
5. Literature review should be strengthened with a focus on the allocation problem. What is the advantage of the proposed approach over the others? I suggest considering other approaches, for example https://www.ein.org.pl/ein/sites/default/files/2021-04-13.pdf or https://www.sciencedirect.com/science/article/pii/S0141933120305159?via%3Dihub
6. The authors should discuss the possibility of expanding the problem to a three-dimensional structure, possibly taking shifts between the shelving walls into account.
7. The middle I/O placement is not very common; why is it the main focus of the research?
8. The study would benefit significantly if the authors discussed the use of two- or four-container (totes) craner
9. "∗" should not be used as a multiplication symbol.
10. Do the authors consider the acceleration and braking characteristics of stacker cranes? They can be of great importance over short distances, sometimes even crucial.
11. Proposition "ready-to-use" simple heuristic guidelines in standard WCS or AS / RS control mechanisms could be a valuable outcome of the research.
